# Mothers’ Emotion Regulation and Negative Affect in Infants: The Role of Self-Efficacy and Knowledge of Parenting Practices

**DOI:** 10.3390/children10010085

**Published:** 2022-12-31

**Authors:** Oriola Hamzallari, Leanna Rosinski, Anton Petrenko, David J. Bridgett

**Affiliations:** 1Department of Psychology, Aleksandër Moisiu University, Rruga Miqësia, Spitallë, 2000 Durrës, Albania; 2Department of Psychology, Northern Illinois University, DeKalb, IL 60115, USA

**Keywords:** parenting suppression, parenting reappraisal, mothers’ self-efficacy, knowledge of parenting practices, infants’ negative affect, early intervention

## Abstract

Early in development, children rely heavily on caregivers for assistance with the regulation of negative emotion. As such, it is important to understand parent characteristics that influence caregiver ability to attenuate infant negative affect and mediating factors by which this process may unfold. This study examined the relationship between parental emotional regulation strategies (ERs) and infants’ negative affect and tested the mediating effects of parenting self-efficacy and knowledge of this association. Results indicated that higher maternal reappraisal was related to higher maternal self-efficacy whereas higher maternal suppression was related to lower knowledge of parenting practices. Maternal suppression was negatively related to infant frustration; maternal self-efficacy was positively related to infant falling reactivity and negatively related to sadness. There was a significant indirect effect between maternal reappraisal and infant falling reactivity through maternal self-efficacy. The mediation result suggests that mothers with higher use of reappraisal show higher self-efficacy and have infants with higher falling reactivity. Maternal knowledge about parenting practices was related to lower infant fear. Maternal knowledge of parenting practices did not mediate any associations between maternal emotion regulation strategies and infant negative affect. These findings contribute to the understanding early protective parenting mechanisms for supporting the external regulation of negative affect in infants and also in designing and implementing preventive parenting programs focused on the emotional needs of parents and children.

## 1. Introduction

Temperament refers to early emerging emotional, regulatory, and attentional characteristics that have a biological basis and undergo rapid development during the early years of life [1]. 6 Environmental factors, including parenting factors, have also shown effects on young children’s temperament [2,3,4]. Existing studies have considered parenting and parent attributes in association with young children’s outcomes, including their temperament (e.g., [5,6,7]). For example, in a study measuring parent emotion regulation strategies (ER) and parenting practices in distressed situations with children, mothers switching of ER strategies without the reduction of negative mood during child distress showed anxiety-related practices that were not supportive of their children’s regulation of anxiety [8]. Findings such as these suggest that parental emotional regulation (ER) is associated with parenting practices (for reviews, see [9,10]) and are of importance in supporting the regulation of children’s affect. The importance of studying parent ER and pathways by which parent ER may influence young children’s affect is highlighted by a recent call for additional work in this area (see [11]).

Self-efficacy is a parent factor that has been related to children’s outcomes. Several studies have examined how low self-efficacy in parents is linked with parents’ higher over-reactivity, less sensitivity, and children’s higher negative affect [2,12,13,14]. However, limited work has considered the link between parent ER and self-efficacy. In one exception, Martini and colleagues [15] suggested that higher reappraisal, a specific ER strategy, and self-efficacy in mothers were important predictors of infants’ quality of sleep, whereas lower self-efficacy and higher suppression, also a specific ER strategy, were linked with higher risk for infant crying, which is often considered a behavioral marker of infant negative affect. Infants that are difficult to soothe may also be a predictor of low self-efficacy in mothers (e.g., [16]). Therefore, studying parents’ emotional coping (e.g., ER strategies) particularly during the early years of children’s development, may lead to a more nuanced understanding of processes parents employ to both manage their reactions to children’s negative affect and that affect their sense of self-efficacy in more difficult parenting situations. 

In sum, limited work has considered ER strategies in parents and their relation to self-efficacy and parental knowledge—parent attributes associated with children’s outcomes. To our knowledge, no work has considered the interplay between these constructs and children’s negative affect in infancy. To address this gap, we examine how maternal ER strategies may predict maternal sense of efficacy and parental knowledge of parenting practices, as well as associations between these maternal ER attributes and infant negative affect. We also consider the possibility that maternal self-efficacy and knowledge of parenting may mediate associations between maternal ER and infant negative affect. The study has important implications because it examines early protective and at-risk emotional parental mechanisms. Identifying relevant parenting factors and potential mechanisms of action as early as possible can support heathy ER in children and lower children’s propensity to experience negative affect, which may be easier to achieve earlier in life before children’s ER and negative affect become more stable [17].

### 1.1. Parental Self-Efficacy and Parenting Knowledge, and Child’s Negative Affect

Infant negative affect may be an early risk factor for behavioral problems in toddlerhood and later in life [18]. However, prior work suggests that parenting can modify the regulation of negative affect early on, potentially reducing risk for these outcomes [4,19]. Parents who use supportive practices, exhibit feelings of competency [20,21], and have more knowledge of child development [22]—factors that may be related to children’s lower negative affect – promote positive developmental trajectories among children. 

Several studies have investigated how parental self-efficacy is linked with children’s negative affect and, relatedly, difficult temperament. On the one hand, studies have revealed that lower parental efficacy is related to poorer regulation of negative affect [15,23]. For example, self-efficacy predicted changes in negative reactivity in infants over time [24]. On the other hand, other studies revealed that mothers with a higher sense of competence who faced difficulties with their children’s negative affect had less conflict and better managed children’s difficult reactions [16,25]. In the longitudinal study by Troutman and colleagues [16], mothers who had children with early high negative emotionality showed lower efficacy, but over the course of the study, these mothers exhibited an increase in self-efficacy that was greater than that of easy infants’ mothers. This increase in sense of competence could be related to coping adaptively with repeated challenging situations of crying children [16]. Because ER strategies are critical to coping in stressful situations, it could be that parents who use more adaptive ER strategies manage difficult child rearing situations, such as displays of infant negative affect, better than parents who experience ER difficulties, resulting in higher efficacy. Several empirical studies support this possibility [26,27,28], which suggest that parent emotional factors are an important influence on parental efficacy. For example, use of reappraisal in parents predicted self-efficacy and children’s use of adaptive emotional regulation strategies [26].

Research is scarcer on associations between parenting knowledge and children’s negative affect. However, some studies suggest that knowledge of child development is associated with positive parenting and less use of negative discipline [29,30]. Dalimonte-Merckling and Brophy-Herb [31] found that parents who had highly reactive children and reported greater knowledge of infant development reported lower child aggression compared to parents who had lower knowledge. These findings demonstrate that parenting knowledge could be an influential predictor of children’s negative affectivity.

Despite the possibility of parental self-efficacy and knowledge being associated with lower infant negative affect, few studies have considered these possibilities, particularly in the first year of life. To address these gaps, the current study considers both maternal self-efficacy and knowledge of parenting in relation to infant negative affect at 12 months of age. 

### 1.2. Parental ER and Negative Affect in Children

Two lines of evidence suggest that parent ER may be related to children’s negative affect. First, parents with better ER may support the development of children’s own ER, which would likely result in fewer child displays of negative affect [32,33,34]. One way this process may unfold is through parental modulation of their own negative affect. Children refer to their parents’ emotional reactions when developing emotional regulation skills, and particularly parent positive emotional expressivity is essential in supporting these processes in young children [35]. In contrast, poorly regulated parental expressions of emotion are associated with distress and maladaptive emotional mechanisms in children [36]. 

Another way that parental ER may be related to children’s negative affect is through the influence of ER on caregiving behaviors. Buckholdt and colleagues [37] revealed that parents reporting higher emotion dysregulation tended to invalidate their children’s emotional expressions, which was related to higher dysregulation for their children ([37]; also see [10] for a review). In support of such findings, more recent studies have shown that maladaptive emotional mechanisms (e.g., rumination, suppression) used by parents contributed to dysregulation of negative affect in young children (e.g., sadness, anger; [38,39]).

All in all, the studies briefly noted above provide evidence of the relevance of parental ER for children’s ER. Additionally, a few studies have taken into consideration how the use of parental ER strategies contributed to the regulation of negative affect in infants and young children. The results from these studies suggest that parental ER is a contributing factor that could play a vital role in parents’ coping strategies when dealing with infants’ and young children’s negative affect and reactivity. 

### 1.3. The Current Study

As outlined above, existing studies point to the links between parent ER and parent self-efficacy and parenting-related skills, including knowledge about parenting [13,15,16,40]. Likewise, in a separate set of studies, these parental attributes have shown relations with children’s negative affect and related attributes, including children’s ER [2,13,23,35,36]. Additional evidence points to associations between parent ER and children’s negative affect [8,18,38,39]. However, to our knowledge, no studies have considered the possibility of parent self-efficacy and knowledge about parenting mediating the influence of parent ER on young children’s negative affect.

To build upon existing studies, in the current study, we used path analysis to examine the associations between maternal ER strategies—specifically, reappraisal and suppression—and maternal parenting self-efficacy and parenting knowledge. We also examined longitudinal associations between these maternal characteristics and specific aspects of infant negative affect. Finally, to extend existing studies, we considered maternal parenting self-efficacy and knowledge about parenting as mediators linking maternal ER to specific aspects of infant negative affect. 

## 2. Methods

### 2.1. Participants

Mother–infant dyads (N = 181) were recruited from a rural community in the Midwestern United States using flyers and referrals from a local OB-GYN. To meet eligibility criteria, infants had to be the result of a full-term pregnancy and not have any developmental concerns, while mothers had to be at least 17 years of age. During the course of the study, two infants developed medical complications and were excluded from analyses, bringing the final sample size to 179. 

The mothers in the study sample were diverse, with 71.51% of the sample being Caucasian, 15.64% Black, 8.94% Hispanic/Latino, 1.10% Native American, and 3.35% “Other.” Mothers’ average age during the first lab visit, which occurred when infants were 4 months old, was 27.49 years (SD = 6.07). Teenage mothers comprised 7.82% of the sample (17–19 years of age), while 12.29% of the sample identified as being single mothers. Average family income was USD 49,718 (SD = USD 38,260). Family poverty status was determined using the income-to-needs ratio, with ratios less than 1.0 indicating that family income falls below the poverty threshold for family size, while a ratio less than 2.0 suggests the family is economically stressed. In this sample, the mean family income-to-needs ratio was 2.15 (SD = 1.67); 25.69% of families reported income falling below the poverty line; 59.22% of families were economically stressed. The average number of years of education completed by mothers was 14.83 (SD = 2.76); 8.94% of the mothers in the sample did not complete high school or receive a certificate of high school diploma equivalency. Nearly half (53.1%) of the participating infants were girls. 

### 2.2. Procedure

Mothers reported on demographic information and completed the Knowledge of Infant Development Inventory [41], which assessed maternal knowledge about parenting, and the Emotion Regulation Questionnaire [42], which assessed two ER strategies, when infants were 4 months of age. Additionally, mothers were interviewed during the 4-month visit using the Structured Clinical Interview for the DSM-IV (SCID-IV; [43]), with depression status incorporated into the cumulative risk index. When infants were 8 months of age, mothers completed the Maternal Self Efficacy Scale [44]. Finally, when children were 12 months of age, mothers reported on their infants’ negative affect using the Infant Behavior Questionnaire-Revised [45].

### 2.3. Measures

*Maternal Knowledge of Parenting Practices.* The Knowledge of Infant Development Inventory (KIDI; [41]) is a 75-item questionnaire which was administered to mothers at 4 months postpartum. Although the KIDI consists of several scales, only the “knowledge of parental practices” scale was used for the purpose of the current study. This scale evaluates the accuracy of parents’ beliefs about parenting and consists of 14 multiple-choice items such as “the more you comfort your crying baby by holding and talking to it, the more you spoil him/her”. The KIDI is a widely used measure of parental awareness of infant development and was developed based on physicians’ input regarding the most crucial aspects of parenting and child development for parents to know [41,46]. The measure has displayed adequate reliability among previous samples [41,46,47]. The parental practices subscale in the current sample showed an acceptable internal consistency of α = 0.60. 

*Maternal Emotion Regulation.* At 4 months postpartum mothers completed the Emotion Regulation Questionnaire (ERQ; [42]). The ERQ is comprised of 10 Likert-type scale items (ranging from 1 = strongly disagree to 7 = strongly agree) regarding one’s use of reappraisal and suppression strategies when attempting to manage emotions. The ERQ has shown good internal consistency and test–retest reliability across samples [42]. In the current sample both scales demonstrated good internal consistency (reappraisal α = 0.80; suppression α = 0.83).

*Maternal Self-Efficacy.* Mothers completed the Maternal Self-Efficacy Scale (MSES; [44]) at 8 months postpartum. The MSES consists of 10 items regarding mothers’ feelings of parenting efficacy. Questions are answered using a Likert-type scale, with response options ranging from 1 (not good at all) to 4 (very good). One example of an item is: “How good do you feel you are at feeding your baby?” The MSES has been shown to be reliable with internal consistency above α = 0.70 across samples [44,48]. Internal consistency in the current sample was good, at α = 0.81.

*Infant Negative Affect.* When infants reached 12 months of age, negative affect was assessed using the Infant Behavior Questionnaire—Revised (IBQ-R), a 184-item parent-report measure devised to evaluate the behavior and temperament of infants 3–12 months of age [45]. Four subscales comprising the Negative Affectivity factor were used in the current study: distress to limitations, fear, sadness, and falling reactivity. Subscales used in the current study have previously shown strong inter-rater reliability among primary and secondary caregivers and high internal consistency above α = 0.80 among items comprising each subscale [45]. In the current sample, internal consistency for each of the four subscales ranged between α = 0.76 and α = 0.91. 

*Cumulative risk.* Mothers filled out a demographic questionnaire and were interviewed using the Structured Clinical Interview for DSM-IV Axis I Disorders [43] during the 4-month visit. The following five risk factors were compiled into a cumulative risk composite: teen motherhood, maternal education less than high school, income-to-needs ratio equal to or less than 1.0, single motherhood, and current or past maternal depression. The presence of each risk factor was assigned a point value of 1 (possible range 0–5), with higher scores indicating the presence of more risk factors.

## 3. Results

### 3.1. Missing Data

By infant age of 12 months when infant temperament was assessed, sample attrition was 20.1%, resulting in missing data. A combined generalized least squares (GLS) test of homogeneity of means and covariances [49] was used to assess the patterns of missing data in the model. The GLS test for the model [χ^2^(180) = 200.96, *p* = 0.135] was non-significant, indicating that the data was likely missing at random. As such, full information likelihood estimation procedures were used to estimate missing values, a procedure which has been shown to result in less biased path estimates and low Type 1 Error rates, and to provide more robust hypothesis testing, compared to listwise deletion [50].

### 3.2. Main Effects

Path modeling was conducted using the statistical software EQS version 6.3 [51]. One model was used to test study questions, with the infant temperament characteristics of frustration, fear, falling reactivity, and sadness as the outcomes. Infant sex and cumulative risk were included as covariates. Zero-order correlations are provided in Table 1. The overall model fit was excellent (χ^2^(3) = 1.45, *p* = 0.69, CFI = 1.00, RMSEA = 0.00, and SRMR = 0.01; See Figure 1) based on standard fit statistic criterion including assessment of comparative fit and absolute fit [52,53].

Direct predictors of maternal self-efficacy and knowledge about parenting were considered first. Results indicated that maternal self-efficacy was predicted by infant sex (*b* * = −0.18, *z* = −2.29, *p* = 0.022); mothers of daughters had higher self-efficacy than mothers with sons. In addition, maternal reappraisal significantly predicted maternal self-efficacy (*b ** = 0.23, *z* = 2.64, *p* = 0.008), whereas suppression exhibited a trend-level association (*b ** = −0.14, *z* = −1.79, *p* = 0.075). Cumulative risk was not significantly related to maternal self-efficacy (*b* * = 0.09, *z* = 1.18, *p* = 0.23). Concerning maternal knowledge of parenting practices, direct predictors included cumulative risk (*b* * = −0.31, *z* = −4.25, *p* < 0.001) and suppression (*b ** = −0.18, *z* = −2.68, *p* = 0.007). Reappraisal displayed a trend-level association with maternal knowledge of parenting (*b ** = 0.12, *z* = 1.72, *p* = 0.085). Infant sex was not directly associated with knowledge of parenting practices (*b ** = −0.01, *z* = −0.09, *p* = 0.93).

Regarding aspects of infant negative affect, maternal suppression emerged as a significant negative predictor of infant frustration (*b ** = −0.19, *z* = −2.27, *p* = 0.023). Other direct predictors of frustration were not significant (Sex: *b* * = 0.04, *z* = 0.44, *p* = 0.66; Cumulative Risk: *b ** = 0.10, *z* = 1.16, *p* = 0.25; Reappraisal: *b ** = 0.10, *z* = 1.22, *p* = 0.22; Self-efficacy: *b* * = −0.16, *z* = −1.51, *p* = 0.13; Knowledge of parenting practices: *b* * = −0.09, *z* = −1.09, *p* = 0.28). Infant fear was directly predicted by cumulative risk (*b ** = 0.26, *z* = 2.95, *p* = 0.003) and knowledge of parenting practices (*b ** = −0.19, *z* = −2.07, *p* = 0.038). Other predictors did not demonstrate significant direct associations with fear (Sex: *b ** = −0.12, *z* = −1.48, *p* = 0.14; Reappraisal: *b ** = 0.11, *z* = 1.17, *p* = 0.24; Suppression: *b ** = −0.10, *z* = −0.92, *p* = 0.36; Self-efficacy: *b ** = 0.06, *z* = 0.66, *p* = 0.51). Infant falling reactivity was directly predicted by cumulative risk (*b ** = −0.22, *z* = −2.55, *p* = 0.011) and maternal self-efficacy (*b ** = 0.25, *z* = 2.79, *p* = 0.005); other predictors were not significantly related (Sex: *b ** = 0.02, *z* = 0.24, *p* = 0.81; Reappraisal: *b ** = 0.01, *z* = 0.06, *p* = 0.95; Suppression: *b ** = 0.10, *z* = 1.13, *p* = 0.26; Knowledge of parenting practices: *b ** = 0.04, *z* = 0.38, *p* = 0.70). Infant sadness was directly predicted by cumulative risk (*b ** = 0.23, *z* = 2.92, *p* = 0.004) and maternal self-efficacy (*b ** = −0.18, *z* = −2.01, *p* = 0.04). No other variables directly predicted sadness (Sex: *b ** = −0.13, *z* = −1.65, *p* = 0.098; Reappraisal: *b ** = −0.03, *z* = −0.36, *p* = 0.72; Suppression: *b ** = −0.07, *z* = −0.68, *p* = 0.49; Knowledge of parenting practices: *b ** = −0.06, *z* = −0.80, *p* = 0.42).

An examination of mediation via measurement of indirect effects showed that reappraisal indirectly predicted infant falling reactivity through maternal self-efficacy (*b ** = 0.06, *z* = 2.08, *p* = 0.037), whereas suppression did not demonstrate a significant indirect effect (*b ** = −0.04, *z* = −1.40, *p* = 0.16). Neither reappraisal (*b ** = −0.01, *z* = −0.33, *p* = 0.74) nor suppression (*b ** = 0.03, *z* = 0.94, *p* = 0.35) indirectly predicted fear. No significant indirect effects via knowledge of parenting practices or maternal self-efficacy were found for infant frustration (Reappraisal: *b ** = −0.05, *z* = −1.49, *p* = 0.13; Suppression: *b ** = 0.04, *z* = 1.54, *p* = 0.12). However, a trend-level indirect effect of reappraisal on sadness was identified (*b ** = 0.05, *z* = −1.73, *p* = 0.08), but not for suppression (*b ** = 0.04, *z* = 1.40, *p* = 0.16).

## 4. Discussion

The present study showed evidence that parental reappraisal was directly related to maternal self-efficacy indicating a potentially important role for this ER strategy in supporting maternal parenting efficacy with infants. This finding supports similar associations from previous studies suggesting that ER factors are influential in parent self-efficacy (e.g., [26]). Reappraisal had no significant association with maternal knowledge of parenting. However, use of suppression among mothers was negatively related to maternal knowledge of parenting. The literature on ER in parents and parental knowledge is scarce and the results of this study contributes to and extends the existing literature by suggesting that maternal ER strategies could have differential effects on parenting related outcomes like self-efficacy and parenting knowledge.

Regarding aspects of infant negative affect, the results showed no direct associations between reappraisal and infant frustration, fear, sadness, or falling reactivity. However, there was an indirect effect between maternal reappraisal and infant falling reactivity—an attribute that, while loading with negative affect during infancy [45], also is indicative of early infant regulatory efforts—through maternal self-efficacy. In addition to maternal parenting self-efficacy showing an association with infant falling reactivity, it also was inversely associated with infant sadness. These findings support previous literature on parenting self-efficacy and young children’s outcomes and negative affect [24,54,55] and emphasize the role of parenting self-efficacy for managing infant expressions of sadness and supporting early infant regulatory efforts that may manifest as falling reactivity. Maternal reappraisal also may be playing a role in these processes in light of the pattern of associations identified in the current study. Specifically, mothers who employ reappraisal may feel more self-efficacy in responding to aspects of infant negative affect (e.g., sadness) and supporting infant regulatory efforts.

The role of maternal suppression in relation to aspects of infant negative affect is less clear than that of reappraisal. To our surprise, higher maternal suppression was related to lower infant frustration. It is possible that mothers employ this ER strategy to suppress outward emotion expressions in the face of infant frustration that may otherwise be dysregulating to infants (e.g., suppressing outward signs of their own frustration in situations where infants are frustrated or angry). Similarly, mothers may employ suppression to help engage in more adaptive caregiving behavior when infants are frustrated, resulting in lower infant frustration. These possibilities are consistent with previous studies indicating that use of suppression may be adaptive when used purposefully, but non-adaptive when used automatically [56,57]. Thus, use of suppression is less predictive in its outcomes compared to reappraisal [58]—a conclusion that may extend to understanding the varied ways in which parents may employ ER strategies in parenting- and family-related contexts. These possibilities will need to be considered in future work that more directly considers possible mechanisms that link maternal suppression to infant frustration.

Knowledge of parenting practices was related to lower infant fearfulness. This suggests that poor knowledge of parenting practices could be a risk factor for fear regulation beginning as early as the first year of life. However, knowledge of parenting practices was not related to other aspects of infant negative affect. Together, these findings suggest that knowledge of parenting more directly contributes to parental ability to respond to displays of infant fear compared to other emotional reactions infants may display. Further studies are needed to directly examine this possibility, as well as specific mechanisms that may link knowledge of parenting to lower infant fearfulness.

Finally, it is worth pointing to several associations between covariates in this study and maternal characteristics and aspects of negative affect. Cumulative risk was associated with maternal knowledge of parenting but not self-efficacy. This may be due to our cumulative risk index being comprised, in part, of indicators of not only economic well-being, but also indicators that may indirectly assess competency in accessing information about parenting (e.g., maternal educational attainment). Higher cumulative risk also was related to higher infant fear and sadness, and lower falling reactivity. It may be that higher cumulative risk within families is associated with higher difficulties of parents to regulate their negative affect and perceptions that infants exhibit more negative affect [15,59,60]. These findings also are consistent with the possibility that stress affects other aspects of family dynamics, such as parenting behavior, which affects children’s temperament [61,62].

### Study Limitations and Future Directions

One limitation of the current study is that parenting practices were not measured. We examined parenting factors that contribute to parenting practices [2,3,63]. It may be that parenting behaviors are affected by maternal self-efficacy and knowledge of parenting, which, in turn, influence infant negative affect. This possibility should be considered in future studies. The current study also did not consider the possibility of child effects. Although several studies have identified children’s temperament effects on parenting factors [64,65], other studies have instead provided strong evidence of effects of parenting factors on children’s outcomes and negative affect [14,16,17,24]. Nevertheless, future research should employ approaches that can account for possible child effects in models that consider maternal ER strategies, self-efficacy, and knowledge of parenting.

Another limitation of this study was the use of only maternal self-report to assess maternal and infant attributes. Future studies should consider using multiple methods to assess attributes of interest (e.g., observations of infant temperament). We also only considered two maternal ER strategies—reappraisal and suppression. Similarly, the measurement of ER did not distinguish between use of adaptive versus non-adaptive reappraisal and suppression [56]. Future studies should consider expanding the types of maternal ER strategies that are considered, and whether strategies are being employed flexibly or rigidly to meet children’s needs.

Despite these limitations this study has contributed to the relatively scarce literature on parent ER, parenting self-efficacy, knowledge of parenting behavior, and infants’ negative affect. Broadly, our findings support the theoretical frameworks of the contributing role of self-regulation and parental ER to parenting-related processes and children’s outcomes [9,10]. Importantly, our findings point to the potential importance of targeting parent emotion regulation for intervention when parenting are struggling with how to respond to children’s negative affect. Likewise, findings support the potential importance of interventions targeting parent self-efficacy and knowledge for improving child and parent outcomes.

## Figures and Tables

**Figure 1 children-10-00085-f001:**
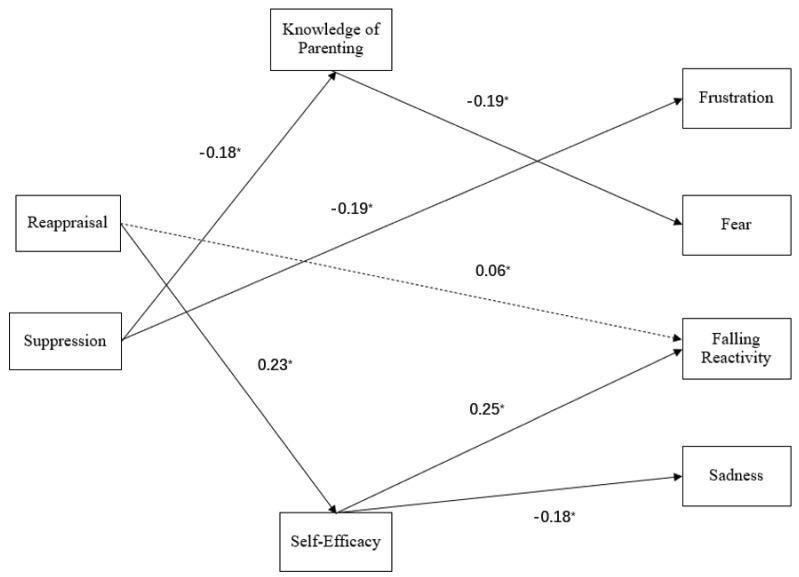
Final path model showing standardized coefficients between all primary variables in the model. Significant direct associations are shown as solid lines. The significant indirect association between maternal reappraisal and infant falling reactivity is shown as a dashed line. Associations between covariates and other variables are not shown for clarity. * *p* < 0.05.

**Table 1 children-10-00085-t001:** Zero-order Correlations Between Variables.

Variables	1	2	3	4	5	6	7	8	9	10
1. Infant Sex										
2. Cumulative Risk	0.06									
3. Maternal Reappraisal	−0.03	−0.10								
4. Maternal Suppression	−0.02	0.10	0.07							
5. Maternal Self-Efficacy	−0.18 *	0.05	0.23 **	−0.10						
6. Knowledge of Parenting Practices	−0.03	−0.33 ***	0.21 **	−0.12	0.11					
7. Infant Frustration	0.07	0.10	0.02	−0.13	−0.15	−0.07				
8. Infant Fear	−0.12	0.28	0.10	−0.03	0.14	−0.2 *	0.27 **			
9. Infant Falling Reactivity	−0.06	−0.20 *	0.09	0.05	0.23 **	0.16	0.41 ***	−0.10		
10. Infant Sadness	−0.06	0.21 *	0.09	−0.02	−0.16	−0.12	0.68 ***	0.37 **	−0.53 ***	

* *p* < 0.05, ** *p* < 0.01, *** *p* < 0.001.

## Data Availability

The data used for this study is available upon request to the first (OH) or last author (DB).

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
