# Peer review of "Mothers’ Emotion Regulation and Negative Affect in Infants: The Role of Self-Efficacy and Knowledge of Parenting Practices"

_children, 2022, doi:10.3390/children10010085_

Round 1

Reviewer 1 Report

1.          The title and figure 1 did not have consistence. I suggest the author should revise the hypothesized model. For example, this study examined the relationship between parental emotional regulation strategies and child’s negative affect, and tested the mediating effects of parenting self-efficacy and knowledge in this relationship. I suggest the author should consider this concern to revise the hypothesized model and organize the manuscript.

2.          Based above, I suggest the author should mainly examine the relationships between parental emotional regulation strategies and child’s negative affect. If necessary, the author could deal with the relationships between these sub-factors. I suggest the author should consider this concern to revise the statistical analysis.

3.          The main topic and interesting focus of this study should be the construct of child’s negative affect. The author should deal with the impacts of parental emotional regulation, parenting self-efficacy and knowledge. I suggest the author should consider this concern to revise the literature review and discussions.

4.          The author employed the SEM to deal with the sample. But the results of this study did not present robust results to confirm the validity and reliability of this measurement. I suggest the author should provide why and what to use the statistical program. The author should follow the mainstream practices based SEM to present the results of the measurement model and structure model.

5.          In the section of the results, I suggest the author should employ the multiple mediating models based on SEM to examine the mediating effects of parenting self-efficacy and knowledge, and show the appropriate table or figure to explain these results.

6.          In the section of the discussion, I suggest the author should focus on how to consider the impacts of parenting self-efficacy and knowledge on child’s negative affect and provide theoretical reflections and practical strategies.

Reviewer 2 Report

I thank the authors for their effort in conducting this research. It is my pleasure to review your manuscript. The research appears to originate and offers important stimuli for both prevention and future research. I propose some minor changes to improve the quality of the manuscript.

- "ERs" appears on line 35. Indicate in the text, before using this abbreviation, what exactly it refers to. After that it is useful to use it.

- Lines 70-87: How could this discrepancy in results be explained?

- I ask if it would be possible to revise the introduction, trying to make it less repetitive and redundant.

Otherwise, the manuscript appears well written, argued and complete throughout.

Therefore, I suggest a minor revision.

Round 2

Reviewer 1 Report

The author mostly incorporated my suggestions and revised the manuscript. The manuscript could be published in this form for the journal.